# Persistence with daily growth hormone among children and adolescents with growth hormone deficiency in Japan

Jane Loftus[1], Jenifer Wogen[2], Darrin Benjumea[2]*, Priti Jhingran[2], Yong Chen[3], Jose Alvir[4], Michael P. Wajnrajch[4,5], Kei Takasawa[6]

1 Pfizer Ltd., Tadworth, Surrey, United Kingdom, 2 Genesis Research, Hoboken, New Jersey, United States of America, 3 Pfizer Inc, Collegeville, Pennsylvania, United States of America, 4 Pfizer Inc, New York, New York, United States of America, 5 New York University Grossman School of Medicine, New York, New York, United States of America, 6 Tokyo Medical and Dental University, Tokyo, Japan

* Darrin.Benjumea@genesisrg.com

## Abstract

### Background

Pediatric growth hormone deficiency (pGHD) is treated with daily somatropin (recombinant human growth hormone) injections. High rates of discontinuation and poor adherence to treatment, which are associated with worse growth outcomes, have been documented previously, for example in the US and Europe. Discontinuation of somatropin has not yet been evaluated using real-world data in Japan.

### Objectives

To describe discontinuation of, and persistence to, daily somatropin treatment among children with pGHD in Japan.

### Methods

This was a retrospective cohort study of children (≥3 and <16 years old) who were prescribed somatropin, using 2 Japan-based databases, Japan Medical Data Center (JMDC) and Medical Data Vision (MDV). Children were required to have ≥1 prescription for somatropin (first prescription = index date) within each study period (1 January 2002–30 September 2021 for JMDC and 1 January 2009–31 October 2021 for MDV) and ≥1 GHD diagnosis code without a somatropin prescription during the 6-months pre-index period. Children were required to be continuously enrolled in the database ≥6 months preceding and ≥3 months following index date. Children were followed for up to 48 months post-index.

Early persistence was defined as the proportion of children with ≥1 refill of somatropin subsequent to the initial prescription. Discontinuation was defined as the first observation of a gap in therapy (using >60, >90, and 120-day gap thresholds)

**Data availability statement:** Data cannot be shared publicly because JDMC and MDV data are licensed datasets. Data are available from the JDMC and MDV for researchers who meet the criteria for access. To learn more about the JMDC database and license the dataset, please follow instructions available on the JMDC website (https://www.jmdc.co.jp/en/jmdc-claims-database/). To learn more about the MDV database and license the dataset, please follow instructions available on MDV website (https://en.mdv.co.jp/). No special access privileges were granted to the authors of this paper when accessing these data.

**Funding:** This study was sponsored by Pfizer. Funders participated in study design, data analysis, decision to publish and preparation of this manuscript. There was no additional external funding received for this study.

**Competing interests:** JL, YC, JA and MPW are all employees of Pfizer Inc. and may hold stock/stock options. Funding from Pfizer Inc., a commercial source, does not alter our adherence to PLOS ONE policies on sharing data and materials. JW, DB, and PJ are employees of Genesis Research, LLC, who were paid consultants to Pfizer in connection with the development of this manuscript. Funding from Pfizer Inc., a commercial source, does not alter our adherence to PLOS ONE policies on sharing data and materials. KT is an employee of Tokyo Medical and Dental University and was an unpaid consultant in connection with the development of this manuscript and has no conflict of interest to report. Somatropin, approved and marketed in Japan, was the product under analysis. This analysis used claims and EHR data to investigate the use of somatropin in Japan. This does not alter our adherence to PLOS ONE policies on sharing data and materials.

between successive somatropin prescription fill dates. Persistence was defined as continuous refills of somatropin with no gaps in therapy. Time to discontinuation/non-persistence was evaluated using Kaplan-Meier methods, and Cox proportional hazards models identified predictors of time to discontinuation.

This analysis utilized de-identified patient data from 2 large, Japanese-based retrospective databases; as such this study does not meet the requirements for institutional review board (IRB) review.

## Results

Among the children included in this study (JMDC N = 452, MDV N = 573), most were male (JMDC 64.8%, MDV 60.0%). Mean age (standard deviation) was 8.8 (3.6) years in JMDC and 7.5 (3.6) years in MDV. Early persistence was high across both cohorts (JMDC 91.2%, MDV 83.4%). Using the 90-day gap definition for discontinuation, a sizable proportion of children discontinued over the follow-up period: JMDC 19% at 12 months, 35% at 48 months; and MDV 33% at 12 months, 54% at 48 months. Fewer discontinuations were observed with the 120-day gap definition (~16% at 48 months in JMDC, ~28% at 48 months in MDV) and more were observed with the 60-day gap definition (~67% at 48 months in JMDC, ~83% at 48 months in MDV). No meaningful predictors of discontinuation were identified.

## Conclusions

Despite high early persistence with somatropin, many children with pGHD in Japan were increasingly non-persistent over time: at 48 months post-index, at least 16% of children discontinued therapy, using the JDMC database and the most conservative measure of gap allowance. These results suggest a need for new strategies to support somatropin medication use over time among children with pGHD in Japan.

## Introduction

Pediatric growth hormone deficiency (pGHD), a rare cause of short stature, affects fewer than 1% of children in Japan. Children with pGHD, are treated with daily subcutaneous injections of recombinant human growth hormone (r-hGH, somatropin), which was licensed in Japan in 1988 [1] commonly between the ages of 3 and 16 years of age. The goal of treatment is to improve the growth rate in childhood to a final adult height within 2 standard deviations of the population mean and to prevent or improve metabolic impairments associated with GHD.

Recent studies of pGHD in the United States (US) have estimated that a sizable proportion of patients in the US discontinue somatropin treatment over time [2]. Many studies have associated good adherence and/or persistence with improved growth outcomes including in the US, United Kingdom (UK), Spain, Turkey, and New Zealand [3–7].

The Japan Medical Data Center (JMDC) database has been used to assess medication adherence and persistence among Japanese children with other chronic health conditions. A study of attention-deficit/hyperactivity disorder (ADHD) medication adherence found mean (standard deviation, SD) adherence using the medication possession ratio (MPR) to be 0.51 (SD: 0.32) [8]. Another study, of adherence to urate-lowering medications for gout and asymptomatic hyperuricemia found median (interquartile range, IQR) MPR 0.70 (IQR: 0.32–0.94) and 0.77 (IQR: 0.32–0.97), respectively [9].

There has been little research on adherence or persistence to somatropin among individuals with pGHD in Japan. However, one study highlighted the importance of copayment support for adolescents and found that among adolescents who were financially supported, 24% discontinued somatropin treatment compared with 48% of those not supported [10]. A recent study of 57 Japanese children with pGHD testing a smartphone app designed to improve somatropin treatment adherence found mean (median) adherence rates of 93% (96%) after 24 weeks [11].

To our knowledge, a study of persistence with somatropin among children with pGHD in Japan has not been conducted. The aim of this study was to describe persistence and discontinuation rates of children diagnosed with pGHD treated with somatropin, using data from 2 databases in Japan.

## Methods

### Data source

This was a retrospective cohort study utilizing 2 Japan-based databases. The JMDC claims database is an epidemiological receipt database that includes inpatient, outpatient, dispensing, and medical examination data from multiple health insurance associations for approximately 7.3 million people (5.9% of population). The Medical Data Vision (MDV) Database is the largest clinical database in Japan, covering 16 million people and approximately 15% of acute hospitals.

### Study population

Children with ≥1 prescription claim for somatropin during the study period (JMDC 01 January 2002–31 September 2021, MDV 01 January 2009–31 October 2021) were included. The date of the first prescription for somatropin during the index period was defined as the index date. Children aged between 3 and <16 years with ≥1 diagnosis code for pGHD in the baseline period 6 months prior to or on the index date were included in the analysis to reflect the common ages for treatment of pGHD. Continuous enrolment in the database was required during the baseline period pre-index date and at least 3 months post-index. Children were excluded from the analysis for prescription claims for somatropin during the baseline period or any other causes/diagnoses associated with short stature including psychosocial dwarfism, celiac disease, primary hypothyroidism, or rickets identified any time prior to the index date.

### Study measures

Study measures (persistence and discontinuation) were evaluated throughout the follow-up period for up to 48 months. Early persistence was defined as patients with >1 refill of somatropin (2 + fills). Discontinuation was defined using three gap definitions, as the first observation of a gap of >60, > 90 or >120 days between successive somatropin prescription fill dates [2,12]. The date of fill of the last claim which occurred at the beginning of the qualifying observed gap was defined as discontinuation date, and the time to discontinuation was calculated as discontinuation date to index date. Persistence was defined as continuous refills of somatropin (no discontinuation, i.e., gaps >60, > 90, > 120 days) during the study follow-up period, as has been done in a previously conducted analysis of somatropin persistence [2,12]. Medication adherence could not be evaluated with the data sources used for this study. Other study variables were also evaluated, including demographics, baseline concomitant medications, baseline mental health diagnoses, anxiety/depression and related mental health disorders, ADHD, and all-cause and GHD-related visits.

## Statistical analysis

Descriptive analyses were performed for all study variables. The mean and standard deviation (SD) were calculated for all continuous variables with frequency counts and percentages calculated for categorical variables. Persistence (non-discontinuation) over time was evaluated using Kaplan-Meier methods to assess time to first discontinuation event. The relationship of patient characteristics with time-to-medication discontinuation was evaluated using Cox proportional hazards models. Hazard ratios with 95% confidence intervals (CI) and p-values were calculated for each of the characteristics.

## Results

### Cohort identification

In the JMDC database 6,913 patients were identified who initiated somatropin treatment between January 2002 and September 2021 (Fig 1). Of these patients, 4,444 were aged ≥3 and <16 years old with a diagnosis of GHD at baseline or in the 6 months prior. Over half of the children (2,429) were excluded due to insufficient database enrolment (6 months before and 3 months after index) and another 1,563 children excluded for other causes/diagnoses associated with short stature. The final study analysis included 452 children with pGHD from the JMDC database.

There were 12,706 patients identified in the MDV database who initiated somatropin treatment between January 2009 and October 2021 (Fig 1). Of these patients, 6,528 were aged ≥3 and <16 years old with a diagnosis of GHD at baseline

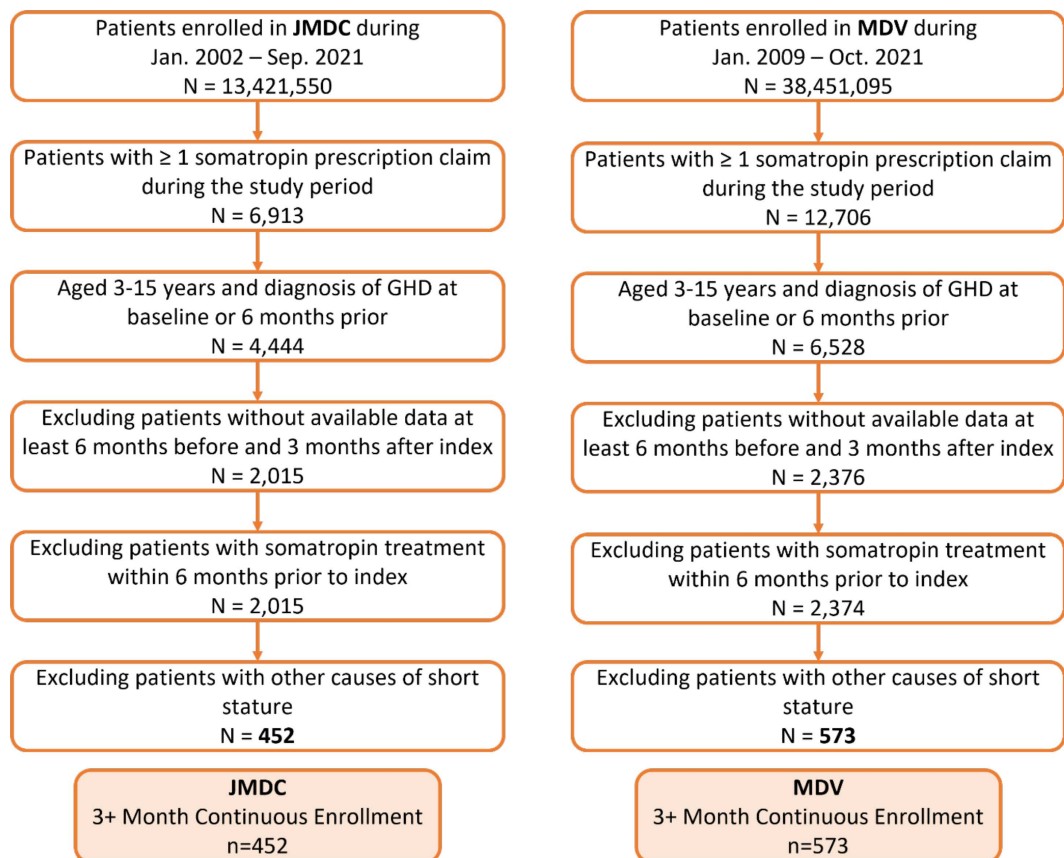

**Fig 1. Cohort sample identification of children (≥3 and <16 years) with pGHD treated with daily somatropin in Japan.** Abbreviations: GHD, growth hormone deficiency; MDV, Medical Data Vision.

or in the 6 months prior. More than half of the children (4,154) were excluded due to insufficient database enrolment (6 months before and 3 months after index) and 1,801 children were excluded for other causes/diagnoses associated with short stature. The final study analysis included 573 children with pGHD from the MDV database.

## Demographics and clinical characteristics

The demographic and clinical characteristics of the JMDC and MDV cohorts are summarized in Table 1. The majority of children in both cohorts were male; JMDC 64.8%, MDV 60.0%. The MDV cohort was younger than the JMDC cohort with mean age for JMDC 8.8 years, MDV 7.5 years and 39.4% of the JMDC cohort were aged 3–7 years compared with 51.8% of the MDV cohort. The majority of children with pGHD were taking at least 3 concomitant medications at index (JMDC: 89.6%; MDV: 67.7%) with a larger percentage of MDV children taking no concomitant medications (JMDC: 4.7%; MDV: 21.3%) compared with either 1 or 2 concomitant medications. Concomitant medications observed were those common to pediatric populations with chronic conditions. Prevalence of other comorbidities was high in both databases, with 95.4% of patients in JMDC and 75.1% of patients in MDV having at least 1 comorbid condition. Comorbid conditions were predominantly respiratory in nature and similar across both databases. A small percentage of both samples were previous childhood cancer survivors (10.0% in JMDC and 6.3% in MDV). Prevalence of anxiety/depression and related mental health disorders was similar across the cohorts (JMDC: 2.7%; MDV: 2.3%) but attention-deficit/hyperactivity disorder (ADHD) was more prevalent in the JMDC cohort (JMDC: 3.1%; MDV: 0.7%).

## Medication persistence and discontinuation

Early persistence (≥1 refill) was very high in both cohorts (JMDC: 91.2%; MDV: 83.4%) with lower early persistence in the MDV cohort in older age categories; 73.4% among children aged 8–11 years and 78% among adolescents aged 12 to <16 years (Fig 2). Mean somatropin medication refill intervals were slightly shorter in the overall JMDC cohort than the MDV cohort (JMDC: 32.6 days; MDV: 33.9 days) and were similar between age groups (Fig 3). Individuals with pGHD in the MDV database were less persistent with somatropin treatment than the JMDC cohort (Fig 4). Using the moderately conservative 90-day gap definition for discontinuation, the percentage of children who continued to be persistent in JMDC was 81% at 12 months and 65% at 48 months and in MDV was 67% at 12 months and 46% at 48 months. Using the 60-day gap definition for discontinuation, lower percentages of children were persistent over the follow-up period; JMDC 55% at 12 months (360 days), 34% at 48 months (1,440 days) and MDV 39% at 12 months, 17% at 48 months. Using the most conservative measure in this study, the 120-day gap definition, more patients were persistent in both the JDMC and MDV database.

In the JMDC patient cohort, there were no significant predictors of persistence/discontinuation identified using the 60- or 90-day gap definition; mental health comorbidities was a significant predictor using the 120-day gap definition and was associated with a lower risk of discontinuation (Table 2). However, in the MDV patient cohort, male sex was associated with lower risk of discontinuation using the 60-day definition and 120-day definition but not using the 90-day definition, in both the unadjusted and adjusted models. Concomitant medication ≥3 was associated with lower risk of discontinuation using the 60- and 90-day definitions in both the unadjusted and adjusted models and one concomitant medication was associated with lower risk in both adjusted and unadjusted models using the 120-day gap definition.

## Discussion

More than 3 in 5 children with GHD treated with somatropin in Japan were male, and the majority were prescribed 3+concomitant medications at baseline. A larger proportion of children from the MDV database were taking no concomitant medications which may be due to the MDV cohort being younger. A higher proportion of patients from the MDV database were young children (aged 3–7 years) and early persistence was high for both data sources. Using a moderately conservative 90-day gap between fill dates to define discontinuation, approximately 19–33% of children with GHD identified in 2 administrative claims databases were non-persistent at 1 year, and 35–54% were non-persistent at 4 years.

**Table 1. Baseline demographic and clinical characteristics of children (≥3 and <16 years) with pGHD treated with daily somatropin in Japan.**

| | JMDC (n = 452) | MDV (n = 573) |
|---|---|---|
| **Sex, n (%)** | | |
| Male | 293 (64.8%) | 344 (60.0%) |
| Female | 159 (35.2%) | 229 (40.0%) |
| **Age [categorical], n (%), mean (SD)** | | |
| Age category 1: 3–7 years | 178 (39.4%), 4.87 (1.37) | 297 (51.8%), 4.44 (1.31) |
| Age category 2: 8–11 years | 143 (31.6%), 9.68 (1.09) | 177 (30.9%), 9.7 (1.08) |
| Age category 3: 12 to <16 years | 131 (29.0%), 13.11 (1.05) | 99 (17.3%), 12.95 (0.95) |
| **Number of concomitant medications, n (%)** | | |
| None | 21 (4.7%) | 122 (21.3%) |
| 1 | 11 (2.4%) | 29 (5.1%) |
| 2 | 15 (3.3%) | 34 (5.9%) |
| 3+ | 405 (89.6%) | 388 (67.7%) |
| **Top 5 concomitant medications by ATC level 3, n (%)†** | | |
| Antihistamines for systemic use | 257 (56.9%) | * |
| Expectorants excluding combinations with antitussives | 248 (54.9%) | 106 (18.5%) |
| Antitussives excluding combinations with expectorants | 193 (42.7%) | * |
| Other analgesics and antipyretics excluding opioids | 166 (36.7%) | * |
| Antithrombotic agents | 158 (35.0%) | 194 (33.9%) |
| Anticholinergic agents | * | 169 (29.5%) |
| Antiadrenergic agents, centrally acting | * | 124 (21.6%) |
| Insulin | * | 90 (15.7%) |
| **Number of patients with at least one comorbid condition, n (%)** | | |
| ≥1 | 431 (95.4%) | 430 (75.1%) |
| **Top 5 comorbid conditions by frequency, n (%)** | | |
| allergic rhinitis | 214 (47.4%) | 86 (15.2%) |
| acute bronchitis | 151 (33.4%) | 53 (9.3%) |
| bronchial asthma | 131 (29.0%) | 87 (15.2%) |
| acute infection of upper respiratory tract | 116 (25.7%) | 50 (8.7%) |
| allergic conjunctivitis | 94 (20.8%) | * |
| constipation | * | 52 (9.1%) |
| **Number of childhood cancer survivors, n (%)** | | |
| Yes | 45 (10.0%) | 36 (6.3%) |
| **Baseline mental health diagnoses, n (%)** | | |
| Anxiety/Depression and Related Mental Health Disorders, n (%) | 12 (2.7%) | 13 (2.3%) |
| Attention-deficit/Hyperactivity Disorder (ADHD), n (%) | 14 (3.1%) | 4 (0.7%) |

*These concomitant medications and comorbidities may be represented in each cohort but are not part of the top 5 most frequent.

Abbreviations: ADHD, attention-deficit/hyperactivity disorder; ATC, anatomical therapeutic chemical; pGHD, pediatric growth hormone deficiency; MDV, Medical Data Vision; n, number; SD, standard deviation.

†Excluding standard solutions and diagnostic tests.

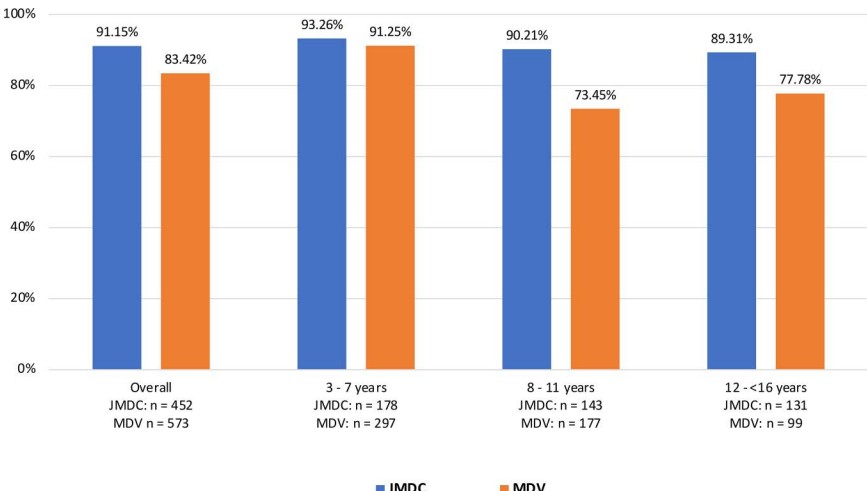

**Fig 2. Early persistence with daily somatropin treatment among children (≥3 and <16 years) with pGHD in Japan.** Percentage of patients with early persistence to daily somatropin, defined as patients with more than one fill (≥1 refill) of a somatropin prescription. Data is represented in the overall cohort and stratified by age category.

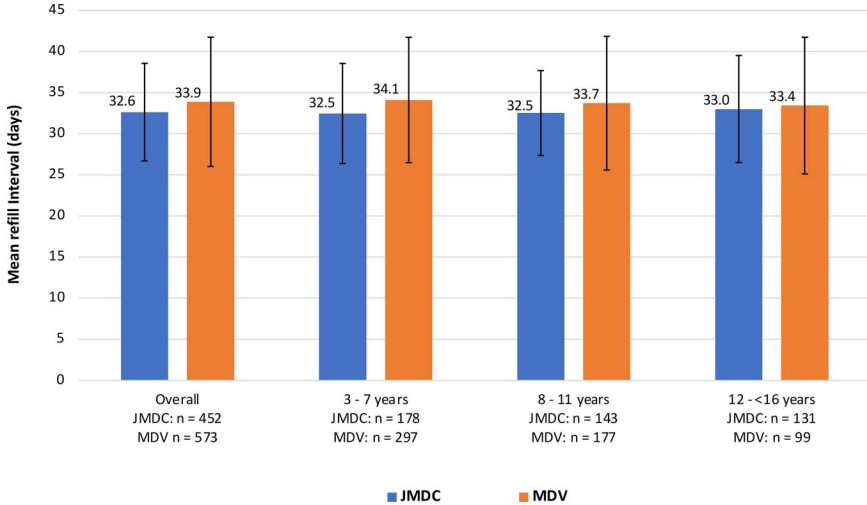

**Fig 3. Somatropin medication refill intervals (days) among children (≥3 and <16 years) with pGHD in Japan.** Mean refill interval in days between subsequent daily somatropin prescriptions. Data is represented in the overall cohort and stratified by age category.

At less than 3%, the prevalence of anxiety/depression and related mental health disorders was low compared to other studies of pGHD populations in the UK (10.3%) and the US (9.8–13.0%) [13]. Attention-deficit conduct and disruptive behavior disorders were more prevalent in the JMDC cohort compared with the MDV cohort, but this may be due to the age distribution of the 2 groups, as the JMDC cohort is older. The prevalence of ADHD in these Japanese cohorts was lower than previous UK-based (6.0%) and US-based (15.0–26.5%) studies [13].

It is unclear how the presence of other comorbidities in this patient population impacts the findings of the study. While studies have rarely assessed the impact of comorbidities on persistence and discontinuation, a few studies have assessed the impact on suboptimal or non-adherence. A recent study conducted in the Australia found that childhood

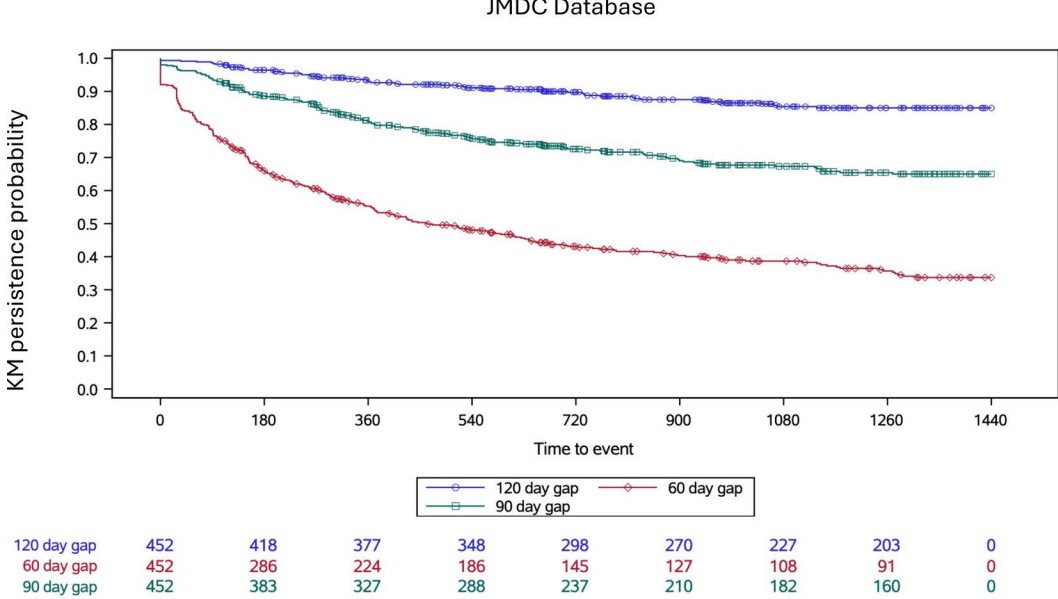

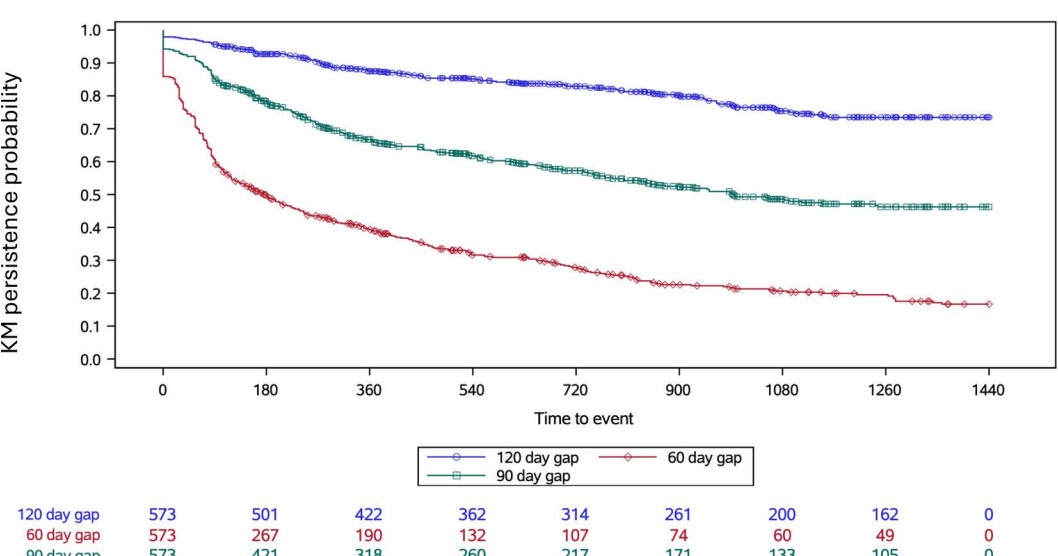

**Fig 4. Kaplan-Meier Analysis of Time to Discontinuation using 60-, 90-Day, and 120-day Gap Definitions in JMDC and MDV Databases.** Kaplan-Meier curves estimating persistence in the JMDC and MDV database. The blue line demonstrates the persistence (or time to discontinuation) to treatment using a 120-day gap definition between subsequent somatropin fills, the green line demonstrates the persistence using a 90-day gap definition, and the red line demonstrates the persistence using a 60-day gap definition. Patients were censored at last available date of data or death, whichever occurred first.

cancer survivors have suboptimal levels of adherence; however, this study assessed adherence to a variety of medical recommendations in adult patients who were childhood cancer survivors [30]. Therefore, the findings of this study are not directly comparable to the findings of our own, although it may suggest a trend in adherence or persistence in this

Table 2. Cox proportional hazard model, 60-, 90-, or 120-day gap allowance.

| Covariate | JMDC | | | | | | MDV | | | | | |
|---|---|---|---|---|---|---|---|---|---|---|---|---|
| | 60-day gap allowance | | 90-day gap allowance | | 120-day gap allowance | | 60-day gap allowance | | 90-day gap allowance | | 120-day gap allowance | |
| | Hazard Ratio (95% CI) | | Hazard Ratio (95% CI) | | Hazard Ratio (95% CI) | | Hazard Ratio (95% CI) | | Hazard Ratio (95% CI) | | Hazard Ratio (95% CI) | |
| | Unadjusted | Adjusted | Unadjusted | Adjusted | Unadjusted | Adjusted | Unadjusted | Adjusted | Unadjusted | Adjusted | Unadjusted | Adjusted |
| **Age: Reference Group: 12 to <16 years** | | | | | | | | | | | | |
| 3 to 7 years | 1.05 (0.78, 1.41) | 1.11 (0.82, 1.52) | 0.87 (0.58, 1.32) | 0.93 (0.61, 1.42) | 0.63 (0.35, 1.16) | 0.66 (0.35, 1.24) | 1.17 (0.89, 1.54) | 1.17 (0.89, 1.55) | 1.07 (0.76, 1.52) | 1.06 (0.74, 1.51) | 0.65 (0.41, 1.04) | 0.63 (0.40, 1.02) |
| 8 to 11 years | 1.08 (0.79, 1.47) | 1.15 (0.83, 1.58) | 0.93 (0.61, 1.41) | 0.98 (0.63, 1.52) | 0.59 (0.31, 1.14) | 0.56 (0.28, 1.12) | 1.21 (0.90, 1.63) | 1.19 (0.89, 1.61) | 1.16 (0.80, 1.68) | 1.11 (0.76, 1.62) | 0.68 (0.41, 1.14) | 0.63 (0.38, 1.07) |
| **Sex: Reference Group: Female** | | | | | | | | | | | | |
| Male | 1.18 (0.92, 1.53) | 1.20 (0.92, 1.55) | 1.20 (0.84, 1.72) | 1.18 (0.81, 1.70) | 1.27 (0.72, 2.24) | 1.10 (0.61, 1.98) | **0.77 (0.64, 0.94)** | **0.79 (0.65, 0.96)** | 0.81 (0.63, 1.03) | 0.82 (0.64, 1.05) | 0.73 (0.51, 1.05) | **0.68 (0.47, 0.98)** |
| **Number of Concomitant Medications: Reference Group: No medications** | | | | | | | | | | | | |
| 1 | 0.58 (0.21, 1.60) | 0.61 (0.22, 1.70) | 0.86 (0.22, 3.34) | 0.84 (0.22, 3.25) | 0.76 (0.24, 2.45) | 0.79 (0.24, 2.59) | 0.66 (0.41, 1.05) | 0.64 (0.40, 1.03) | 0.66 (0.36, 1.20) | 0.65 (0.36, 1.18) | **0.54 (0.36, 0.81)** | **0.56 (0.37, 0.84)** |
| 2 | 0.62 (0.26, 1.47) | 0.62 (0.26, 1.48) | 0.76 (0.22, 2.58) | 0.69 (0.20, 2.38) | 0.90 (0.15, 5.36) | 0.69 (0.11, 4.24) | 0.82 (0.53, 1.25) | 0.83 (0.54, 1.28) | 0.73 (0.42, 1.26) | 0.73 (0.42, 1.27) | 0.61 (0.27, 1.38) | 0.58 (0.26, 1.31) |
| ≥3 | 0.63 (0.37, 1.09) | 0.62 (0.36, 1.07) | 0.80 (0.37, 1.72) | 0.79 (0.37, 1.71) | 0.62 (0.06, 5.94) | 0.56 (0.06, 5.39) | **0.66 (0.52, 0.84)** | **0.66 (0.52, 0.84)** | **0.63 (0.47, 0.84)** | **0.63 (0.47, 0.84)** | 0.51 (0.20, 1.31) | 0.54 (0.21, 1.37) |
| **Mental Health Comorbidities: Reference Group: No Mental Health Comorbidity** | | | | | | | | | | | | |
| Mental Health Comorbidity: Yes | 1.20 (0.72, 1.98) | 1.21 (0.73, 2.02) | 1.57 (0.82, 2.99) | 1.54 (0.84, 2.96) | 2.12 (0.91, 4.93) | **0.06 (0.00, 0.89)** | 1.04 (0.60, 1.80) | 1.03 (0.59, 1.80) | 1.24 (0.64, 2.42) | 1.22 (0.63, 2.40) | 0.51 (0.13, 2.05) | 0.55 (0.13, 2.23) |

Abbreviations: CI, confidence interval.

Bolded hazard ratios and 95% CIs represent statistically significant results (p-value < 0.05).

population. In this study, however, childhood cancer survivor was not assessed as a predictor of persistence. Mental health disorders including anxiety/depression and ADHD are a predictor of non-adherence to medication in children and adolescents [14]. A previous meta-analysis of use of psychotropic medications in adolescents aged 12–18 found a median non-adherence rate of 33% [15]. Among the included studies a positive correlation was found between affective disorders (anxiety/depression) and non-adherence [16,17], as well as ADHD and non-adherence [18]. One study found that adolescents with medications for both depression and ADHD were less likely to be non-compliant [19]. In our study the majority of children with pGHD were taking at least 3 concomitant medications at index, with concomitant use of medication ≥3 being a predictor for lower risk of discontinuation in the MDV cohort. In a survey of caregivers responsible for daily delivery of GH to children aged 14 and younger in Japan, common reasons for non-adherence in the poorly adherent cohort included concerns about safety and side effects [20]. Future studies could further explore the possibility that the positive association between number of concomitant medications and persistence and adherence.

While prevalence of anxiety/depression and related mental health disorders was similar across the cohorts, prevalence of ADHD was more than 3 times greater in the JMDC cohort. In a retrospective study of Japanese healthcare databases, prevalence rate of ADHD in children increased from 0.54% to 1.27% in boys and from 0.10% to 0.32% in girls between 2012 and 2018 [21]. Among Japanese children and adolescents, the highest prevalence of ADHD medication usage is among children aged 7–12. Additionally, adherence rates for ADHD medications are greater for children aged 7–12 (65%) versus adolescents aged 16–18 (43%) [22]. In general, a greater sense of autonomy with age has been suggested as a risk factor for increased non-compliance in adolescents versus children [23]. In contrast, while our study found that individuals with pGHD in the MDV database, which was younger overall, were less persistent with somatropin treatment than the JMDC cohort, age was not found to be a significant predictor of compliance in either cohort.

We found that high proportions of children discontinued somatropin, and discontinuation rates increased with greater duration of available follow-up, consistent with results from other studies [2]. Using the more conservative 90-day gap definition for discontinuation, a sizable proportion of children discontinued over the follow-up period. Using the most conservative 120-day gap definition, fewer patients discontinued but there was still approximately 16% of patients who discontinued at 48 months in the JDMC cohort and 28% at 48 months in the MDV cohort.

Persistence with somatropin may impact the probability of children with GHD achieving final target height. For example, one US-based study found that children who had achieved final height were on somatropin longer (46 ± 21 months, n = 288) than those who discontinued [7]. After 3 years of follow up, children who had achieved final height had greater corrected height standard deviation scores [−0.3 ± 0.90 (n = 179)] and were more persistent with somatropin than those who discontinued [7].

While fewer studies have evaluated the relationship between persistence to somatropin among children with GHD and change in height velocity, many studies have found an association between suboptimal adherence with somatropin treatment and poor growth outcomes. Suboptimal adherence to somatropin daily injections has been shown to lead to poorer growth outcomes [3,24,25]. A New Zealand based study, using rate of returned vials, found that children with good compliance (defined as no more than one missed dose per week) had significantly greater linear growth [3]. A retrospective cohort study of US claims data showed adherent children with pGHD grew an additional 1.8 cm over 1 year of treatment compared with non-adherent children [6]. A UK-based study of 75 children with pGHD who attended regional pediatric endocrinology clinics and found that almost 1 in 4 children (23%) missed >2 injections per week and this was associated with lower predicted height velocities [5]. Finally, another UK-based study of 52 children found that year-on-year height standard deviation score was significantly increased for children who were adherent (defined as proportion of days covered >0.80) to treatment over 3 years, but there was no significant increase for children who were non-adherent [26].

In this study, no meaningful predictors of persistence were identified via Cox proportional hazards regression in the JMDC patient cohort and only 2 meaningful predictors (sex for the 60-day definition and concomitant medications for both definitions) in the MDV patient cohort. Previous research has highlighted predictors of persistence

with somatropin treatment. A recent study using the Easypod autoinjector found indicators associated with better persistence included at least 1 dose change per year, starting treatment at an early age, high adherence (≥85%), and customized injection speed setting; males also had better persistence than females [27]. The same study found variations in median persistence across different regions: 1.0 years among patients in Asia-Pacific regions, 1.5 years among patients in North America and 2.8 years among patients in Europe [27]. Several previous studies have found age to be a predictor of somatropin adherence [3,28–30]. Our study did not find age to be a significant predictor in either MDV or JMDC.

Previous research has found copayment support to be an important factor for somatropin treatment persistence in Japan; among adolescents who were financially supported, 24% discontinued somatropin treatment compared with 48% of those not supported [10]. In Japan, patients with some rare diseases, including pGHD, have access to financial support for out-of-pocket payments via national and local government programs [10]. The Medical Aid Program for Chronic Pediatric Diseases of Specified Categories (MAPChD), the main program used to support children with pGHD, covers all patients in Japan, but eligibility and assistance available from local programs varies by municipality. The present study was unable to assess the impact of financial support on somatropin persistence due to limitations of the JMDC and MDV databases, however the lack of consistent financial support across the Japanese healthcare system could have impacted this study's findings.

## Limitations

This was a retrospective study, and as such, the data were not collected for research purposes and missing data may have resulted in bias. Neither JMDC nor MDV collect information regarding days supply of prescription for injectable treatments such as somatropin, and therefore this study was unable to assess medication adherence. The databases also do not include reasons for discontinuation. JMDC is a claims database including inpatient, outpatient, and dispensing information, whereas MDV is specifically a hospital-based claims database and therefore lacks continuous data from other facilities, such as medical and prescriptions histories, which may explain the lower persistence results among the MDV cohort and the lower percentages of comorbidities and concomitant medications reported compared to JMDC. It was outside the authors ability to assess the extent of patient overlap between the 2 databases. It is for this reason that to the extent possible, the results from the 2 databases have been presented separately in this publication.

Patients were selected into the study cohort based on receipt of a somatropin prescription and a diagnosis of GHD. Patients were excluded if evidence of other causes of short stature were noted in the data, resulting in an attrition of approximately 75% of the study population. In previously published studies in this patient population in the US, a similar exclusion criterion resulted in close to 50% of cohort attrition [2]. The increase in cohort attrition in this study may be a result of varying coding practices between the regions and lower specificity of claims codes in the Japanese system compared to the ICD-10 coding system. This exclusion criterion was included in an attempt to limit potential confounding related to the presence of other conditions that may cause short stature and lead to a somatropin prescription. However, it is possible that patients with pGHD may also have comorbid conditions that contribute to short stature, in which case these patients would have been excluded. Therefore, this study may represent an underestimation of the patients with pGHD in these Japanese databases. It is unclear how this limitation would impact discontinuation and persistence to somatropin.

Finally, the cut-offs for discontinuation and persistence used in this study (60- and 90-day prescription gaps) represent those used in previous studies on this topic [2,12]. In chronic conditions, where visits to the physician's office may be less frequent, the 90-day discontinuation assumption may be more reflective of clinical practice. For this reason, the 90-day cut-off has been emphasized in the results of this publication. Other publications assessing persistence in growth hormone injection therapy have suggested that an appropriate cut-off for discontinuation is a 6-month gap [27,31]. Depending on the specific clinical practices across physicians in Japan, these discontinuation thresholds may lead to either an under

or over-estimation of non-persistence to therapy, as no standard consensus definition of persistence and discontinuation for growth hormone therapy in pGHD exists.

## Conclusions

This evaluation of a real-world pGHD population in Japan found high levels of discontinuation, regardless of patient age. Additionally, persistence with somatropin decreased as duration of follow up increased. As a strong link has been established between persistence and treatment success, strategies to improve treatment persistence among children with pGHD may improve clinical outcomes. Additional research is warranted among different patient populations and larger study samples to quantify non-persistence and suboptimal adherence to somatropin and further characterize potential impact on growth outcomes.

## Acknowledgments

Medical writing support was provided by Amy Glenwright of Genesis Research, LLC.

## Author contributions

**Conceptualization:** Jane Loftus, Darrin Benjumea, Yong Chen, Jose Alvir, Michael P. Wajnrajch, Kei Takasawa.

**Data curation:** Yong Chen, Jose Alvir.

**Formal analysis:** Jenifer Wogen, Yong Chen, Jose Alvir.

**Funding acquisition:** Jane Loftus, Michael P. Wajnrajch.

**Investigation:** Jane Loftus, Priti Jhingran, Yong Chen, Jose Alvir, Michael P. Wajnrajch, Kei Takasawa.

**Methodology:** Jane Loftus, Jenifer Wogen, Darrin Benjumea, Priti Jhingran, Yong Chen, Jose Alvir, Kei Takasawa.

**Project administration:** Jane Loftus, Darrin Benjumea, Priti Jhingran, Michael P. Wajnrajch.

**Resources:** Jane Loftus, Michael P. Wajnrajch.

**Supervision:** Jane Loftus, Priti Jhingran, Michael P. Wajnrajch.

**Validation:** Jane Loftus, Jenifer Wogen, Darrin Benjumea, Yong Chen, Jose Alvir, Kei Takasawa.

**Visualization:** Jenifer Wogen, Darrin Benjumea, Yong Chen, Jose Alvir, Kei Takasawa.

**Writing – original draft:** Jenifer Wogen, Darrin Benjumea.

**Writing – review & editing:** Jane Loftus, Darrin Benjumea, Priti Jhingran, Yong Chen, Jose Alvir, Michael P. Wajnrajch, Kei Takasawa.

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
