## [Decision Letter · Decision Letter 0]

30 Apr 2025

Persistence with daily growth hormone among children and adolescents with growth hormone deficiency in Japan

PONE-D-24-37032

Dear Dr. Benjumea,

We’re pleased to inform you that your manuscript has been judged scientifically suitable for publication and will be formally accepted for publication once it meets all outstanding technical requirements.

Kind regards,

Patrick Goymer

Staff Editor

PLOS ONE

Reviewers' comments:

Reviewer's Responses to Questions

**Comments to the Author**

1. Is the manuscript technically sound, and do the data support the conclusions?

Reviewer #1: Yes

2. Has the statistical analysis been performed appropriately and rigorously?

Reviewer #1: Yes

3. Have the authors made all data underlying the findings in their manuscript fully available?

Reviewer #1: Yes

4. Is the manuscript presented in an intelligible fashion and written in standard English?

Reviewer #1: Yes

Reviewer #1: Loftus and colleagues present a retrospective cohort study of somatropin persistence in Japanese children with pGHD. Here, the authors utilized two Japanese databases to investigate somatropin persistence from 2002 to 2021. The authors provide evidence that discontinuation rates ranged from 16% to 54% (depending on the criteria used) over a 48-month follow-up period. This is a well-written manuscript that appears to have undergone several rounds of revision previously. I believe this manuscript is suitable for publication.

Comments to the Author:

1. Is the manuscript technically sound, and do the data support the conclusions?

Yes, this study employed an adequate sample size and description of the study population, as well as the databases from which the data were derived. The data presented are consistent with the authors’ conclusions.

2. Has the statistical analysis been performed appropriately and rigorously?

Yes, the authors describe how descriptive statistics were computed for continuous and frequency variables. The authors also detail how time-to-event data and potential relationships between patient characteristics and discontinuation were analyzed with Kaplan-Meier and Cox proportional hazard analyses, respectively.

3. Have the authors made all data underlying the findings in their manuscript fully available?

Yes, the authors describe the licensed databases where data were collected.

4. Is the manuscript presented in an intelligible fashion and written in standard English?

Yes, this manuscript is well presented. It should be noted that in several places (lines 123, 129, 143, 167, and 171) an “error” appears where a figure is presumably referenced in the main text. This should be corrected in the publication process.

**Do you want your identity to be public for this peer review?** For information about this choice, including consent withdrawal, please see our Privacy Policy

Reviewer #1: No

---

## [Editor Report · Acceptance letter]

PONE-D-24-37032

PLOS ONE

Dear Dr. Benjumea,

I'm pleased to inform you that your manuscript has been deemed suitable for publication in PLOS ONE. Congratulations! Your manuscript is now being handed over to our production team.

Kind regards,

on behalf of

Dr Patrick Goymer

Staff Editor

PLOS ONE